# Mixture-Rank Matrix Approximation for Collaborative Filtering

**Dongsheng Li**[1]   **Chao Chen**[1]   **Wei Liu**[2*]   **Tun Lu**[3,4]   **Ning Gu**[3,4]   **Stephen M. Chu**[1]

[1]IBM Research - China
[2]Tencent AI Lab, China
[3]School of Computer Science, Fudan University, China
[4]Shanghai Key Laboratory of Data Science, Fudan University, China
{ldsli, cshchen, schu}@cn.ibm.com, wliu@ee.columbia.edu, {lutun, ninggu}@fudan.edu.cn

## Abstract

Low-rank matrix approximation (LRMA) methods have achieved excellent accuracy among today's collaborative filtering (CF) methods. In existing LRMA methods, the rank of user/item feature matrices is typically fixed, *i.e.*, the same rank is adopted to describe all users/items. However, our studies show that submatrices with different ranks could coexist in the same user-item rating matrix, so that approximations with fixed ranks cannot perfectly describe the internal structures of the rating matrix, therefore leading to inferior recommendation accuracy. In this paper, a mixture-rank matrix approximation (MRMA) method is proposed, in which user-item ratings can be characterized by a mixture of LRMA models with different ranks. Meanwhile, a learning algorithm capitalizing on iterated condition modes is proposed to tackle the non-convex optimization problem pertaining to MRMA. Experimental studies on MovieLens and Netflix datasets demonstrate that MRMA can outperform six state-of-the-art LRMA-based CF methods in terms of recommendation accuracy.

## 1 Introduction

Low-rank matrix approximation (LRMA) is one of the most popular methods in today's collaborative filtering (CF) methods due to high accuracy [11, 12, 13, 17]. Given a targeted user-item rating matrix $R \in \mathbb{R}^{m \times n}$, the general goal of LRMA is to find two rank-$k$ matrices $U \in \mathbb{R}^{m \times k}$ and $V \in \mathbb{R}^{n \times k}$ such that $R \approx \hat{R} = UV^T$. After obtaining the user and item feature matrices, the recommendation score of the $i$-th user on the $j$-th item can be obtained by the dot product between their corresponding feature vectors, *i.e.*, $U_i V_j^T$.

In existing LRMA methods [12, 13, 17], the rank $k$ is considered fixed, *i.e.*, the same rank is adopted to describe all users and items. However, in many real-world user-item rating matrices, *e.g.*, Movielens and Netflix, users/items have a significantly varying number of ratings, so that submatrices with different ranks could coexist. For instance, a submatrix containing users and items with few ratings should be of a low rank, *e.g.*, 10 or 20, and a submatrix containing users and items with many ratings may be of a relatively higher rank, *e.g.*, 50 or 100. Adopting a fixed rank for all users and items cannot perfectly model the internal structures of the rating matrix, which will lead to imperfect approximations as well as degraded recommendation accuracy.

In this paper, we propose a mixture-rank matrix approximation (MRMA) method, in which user-item ratings are represented by a mixture of LRMA models with different ranks. For each user/item, a probability distribution with a Laplacian prior is exploited to describe its relationship with different

LRMA models, while a joint distribution of user-item pairs is employed to describe the relationship between the user-item ratings and different LRMA models. To cope with the non-convex optimization problem associated with MRMA, a learning algorithm capitalizing on iterated condition modes (ICM) [1] is proposed, which can obtain a local maximum of the joint probability by iteratively maximizing the probability of each variable conditioned on the rest. Finally, we evaluate the proposed MRMA method on Movielens and Netflix datasets. The experimental results show that MRMA can achieve better accuracy compared against state-of-the-art LRMA-based CF methods, further boosting the performance for recommender systems leveraging matrix approximation.

## 2   Related Work

Low-rank matrix approximation methods have been leveraged by much recent work to achieve accurate collaborative filtering, *e.g.*, PMF [17], BPMF [16], APG [19], GSMF [20], SMA [13], *etc.* These methods train one user feature matrix and one item feature matrix first and use these feature matrices for all users and items without any adaptation. However, all these methods adopt fixed rank values for the targeted user-item rating matrices. Therefore, as analyzed in this paper, submatrices with different ranks could coexist in the rating matrices and only adopting a fixed rank cannot achieve optimal matrix approximation. Besides stand-alone matrix approximation methods, ensemble methods, *e.g.*, DFC [15], LLORMA [12], WEMAREC [5], *etc.*, and mixture models, *e.g.*, MPMA [4], *etc.*, have been proposed to improve the recommendation accuracy and/or scalability by weighing different base models across different users/items. However, the above methods do not consider using different ranks to derive different base models. In addition, it is desirable to borrow the idea of mixture-rank matrix approximation (MRMA) to generate more accurate base models in the above methods and further enhance their accuracy.

In many matrix approximation-based collaborative filtering methods, auxiliary information, *e.g.*, implicit feedback [9], social information [14], contextual information [10], *etc.*, is introduced to improve the recommendation quality of pure matrix approximation methods. The idea of MRMA is orthogonal to these methods, and can thus be employed by these methods to further improve their recommendation accuracy. In general low-rank matrix approximation methods, it is non-trivial to directly determine the maximum rank of a targeted matrix [2, 3]. Candès et al. [3] proved that a non-convex rank minimization problem can be equivalently transformed into a convex nuclear norm minimization problem. Based on this finding, we can easily determine the range of ranks for MRMA and choose different $K$ values (the maximum rank in MRMA) for different datasets.

## 3   Problem Formulation

In this paper, upper case letters such as $R, U, V$ denote matrices, and $k$ denotes the rank for matrix approximation. For a targeted user-item rating matrix $R \in \mathbb{R}^{m \times n}$, $m$ denotes the number of users, $n$ denotes the number of items, and $R_{i,j}$ denotes the rating of the $i$-th user on the $j$-th item. $\hat{R}$ denotes the low-rank approximation of $R$. The general goal of $k$-rank matrix approximation is to determine user and item feature matrices, *i.e.*, $U \in \mathbb{R}^{m \times k}, V \in \mathbb{R}^{n \times k}$, such that $R \approx \hat{R} = UV^T$. The rank $k$ is considered low, because $k \ll \min\{m, n\}$ can achieve good performance in many CF applications.

In real-world rating matrices, *e.g.*, Movielens and Netflix, users/items have a varying number of ratings, so that a lower rank which best describes users/items with less ratings will easily underfit the users/items with more ratings, and similarly a higher rank will easily overfit the users/items with less ratings. A case study is conducted on the Movielens (1M) dataset (with 1M ratings from 6,000 users on 4,000 movies), which confirms that internal submatrices with different ranks indeed coexist in the rating matrix. Here, we run the probabilistic matrix factorization (PMF) method [17] using $k = 5$ and $k = 50$, and then compare the root mean square errors (RMSEs) for the users/items with less than 10 ratings and more than 50 ratings.

As shown in Table 1, when the rank is 5, the users/items with less than 10 ratings achieve lower RMSEs than the cases when the rank is 50. This indicates that the PMF model overfits the users/items with less than 10 ratings when $k = 50$. Similarly, we can conclude that the PMF model underfits the users/items with more than 50 ratings when $k = 5$. Moreover, PMF with $k = 50$ achieves lower RMSE (higher accuracy) than PMF with $k = 5$, but the improvement comes with sacrificed accuracy for the users and items with a small number of ratings, *e.g.*, less than 10. This study shows that PMF

Table 1: The root mean square errors (RMSEs) of PMF [17] for users/items with different numbers of ratings when rank $k = 5$ and $k = 50$.

|  | rank $= 5$ | rank $= 50$ |
| --- | --- | --- |
| #user ratings $< 10$ | 0.9058 | 0.9165 |
| #user ratings $> 50$ | 0.8416 | 0.8352 |
| #item ratings $< 10$ | 0.9338 | 0.9598 |
| #item ratings $> 50$ | 0.8520 | 0.8418 |
| All | 0.8614 | 0.8583 |

with fixed rank values cannot perfectly model the internal mixture-rank structure of the rating matrix. To this end, it is desirable to model users and items with different ranks.

## 4 Mixture-Rank Matrix Approximation (MRMA)

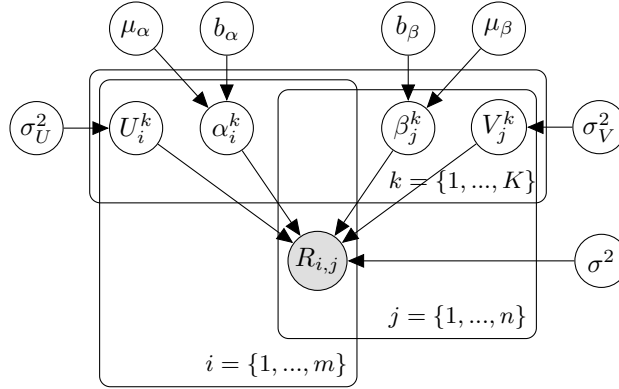

Figure 1: The graphical model for the proposed mixture-rank matrix approximation (MRMA) method.

Following the idea of PMF, we exploit a probabilistic model with Gaussian noise to model the ratings [17]. As shown in Figure 1, the conditional distribution over the observed ratings for the mixture-rank model can be defined as follows:

$$\Pr(R|U, V, \alpha, \beta, \sigma^2) = \prod_{i=1}^{m} \prod_{j=1}^{n} [\sum_{k=1}^{K} \alpha_i^k \beta_j^k \mathcal{N}(R_{i,j}|U_i^k V_j^{k^T}, \sigma^2)]^{\mathbb{1}_{i,j}}, \tag{1}$$

where $\mathcal{N}(x|\mu, \sigma^2)$ denotes the probability density function of a Gaussian distribution with mean $\mu$ and variance $\sigma^2$. $K$ is the maximum rank among all internal structures of the user-item rating matrix. $\alpha^k$ and $\beta^k$ are the weight vectors of the rank-$k$ matrix approximation model for all users and items, respectively. Thus, $\alpha_i^k$ and $\beta_j^k$ denote the weights of the rank-$k$ model for the $i$-th user and $j$-th item, respectively. $U^k$ and $V^k$ are the feature matrices of the rank-$k$ matrix approximation model for all users and items, respectively. Likewise, $U_i^k$ and $V_j^k$ denote the feature vectors of the rank-$k$ model for the $i$-th user and $j$-th item, respectively. $\mathbb{1}_{i,j}$ is an indication function, which will be 1 if $R_{i,j}$ is observed and 0 otherwise.

By placing a zero mean isotropic Gaussian prior [6, 17] on the user and item feature vectors, we have

$$\Pr(U^k|\sigma_U^2) = \prod_{i=1}^{m} \mathcal{N}(U_i^k|0, \sigma_U^2 I), \quad \Pr(V^k|\sigma_V^2) = \prod_{j=1}^{n} \mathcal{N}(V_j^k|0, \sigma_V^2 I). \tag{2}$$

For $\alpha^k$ and $\beta^k$, we choose a Laplacian prior here, because the models with most suitable ranks for each user/item should be with large weights, *i.e.*, $\alpha^k$ and $\beta^k$ should be sparse. By placing the Laplacian prior on the user and item weight vectors, we have

$$\Pr(\alpha^k|\mu_\alpha, b_\alpha) = \prod_{i=1}^{m} \mathcal{L}(\alpha_i^k|\mu_\alpha, b_\alpha), \quad \Pr(\beta^k|\mu_\beta, b_\beta) = \prod_{j=1}^{n} \mathcal{L}(\beta_j^k|\mu_\beta, b_\beta), \tag{3}$$

where $\mu_\alpha$ and $b_\alpha$ are the location parameter and scale parameter of the Laplacian distribution for $\alpha$, respectively, and accordingly $\mu_\beta$ and $b_\beta$ are the location parameter and scale parameter for $\beta$.

The log of the posterior distribution over the user and item features and weights can be given as follows:

$$
\begin{aligned}
l & = \ln \Pr(U, V, \alpha, \beta | R, \sigma^2, \sigma_U^2, \sigma_V^2, \mu_\alpha, b_\alpha, \mu_\beta, b_\beta) \\
& \propto \ln \big[ \Pr(R|U, V, \alpha, \beta, \sigma^2) \Pr(U|\sigma_U^2) \Pr(V|\sigma_V^2) \Pr(\alpha|\mu_\alpha, b_\alpha) \Pr(\beta|\mu_\beta, b_\beta) \big] \\
& = \sum_{i=1}^{m} \sum_{j=1}^{n} \mathbb{1}_{i,j} \big[ \ln \sum_{k=1}^{K} \alpha_i^k \beta_j^k \mathcal{N}(R_{i,j} | U_i^k (V_j^k)^T, \sigma^2 I) \big] \\
& \quad - \frac{1}{2\sigma_U^2} \sum_{k=1}^{K} \sum_{i=1}^{m} (U_i^k)^2 - \frac{1}{2\sigma_V^2} \sum_{k=1}^{K} \sum_{j=1}^{n} (V_i^k)^2 - \frac{1}{2} Km \ln \sigma_U^2 - \frac{1}{2} Kn \ln \sigma_V^2 \quad (4) \\
& \quad - \frac{1}{b_\alpha} \sum_{k=1}^{K} \sum_{i=1}^{m} |\alpha_i^k - \mu_\alpha| - \frac{1}{b_\beta} \sum_{k=1}^{K} \sum_{j=1}^{n} |\beta_j^k - \mu_\beta| - \frac{1}{2} \sum_{k=1}^{K} m \ln b_\alpha^2 - \frac{1}{2} \sum_{k=1}^{K} n \ln b_\beta^2 + C,
\end{aligned}
$$

where $C$ is a constant that does not depend on any parameters. Since the above optimization problem is difficult to solve directly, we obtain its lower bound using Jensen's inequality and then optimize the following lower bound:

$$
\begin{aligned}
l' & = -\frac{1}{2\sigma^2} \sum_{i=1}^{m} \sum_{j=1}^{n} \mathbb{1}_{i,j} \big[ \sum_{k=1}^{K} \alpha_i^k \beta_j^k (R_{i,j} - U_i^k (V_j^k)^T)^2 \big] - \frac{1}{2} \sum_{i=1}^{m} \sum_{j=1}^{n} \mathbb{1}_{i,j} \ln \sigma^2 \\
& \quad - \frac{1}{2\sigma_U^2} \sum_{k=1}^{K} \sum_{i=1}^{m} (U_i^k)^2 - \frac{1}{2\sigma_V^2} \sum_{k=1}^{K} \sum_{j=1}^{n} (V_i^k)^2 - \frac{1}{2} Km \ln \sigma_U^2 - \frac{1}{2} Kn \ln \sigma_V^2 \quad (5) \\
& \quad - \frac{1}{b_\alpha} \sum_{k=1}^{K} \sum_{i=1}^{m} |\alpha_i^k - \mu_\alpha| - \frac{1}{b_\beta} \sum_{k=1}^{K} \sum_{j=1}^{n} |\beta_j^k - \mu_\beta| - \frac{1}{2} Km \ln b_\alpha^2 - \frac{1}{2} Kn \ln b_\beta^2 + C.
\end{aligned}
$$

If we keep the hyperparameters of the prior distributions fixed, then maximizing $l'$ is similar to the popular least square error minimization with $\ell_2$ regularization on $U$ and $V$ and $\ell_1$ regularization on $\alpha$ and $\beta$. However, keeping the hyperparameters fixed may easily lead to overfitting because MRMA models have many parameters.

## 5 Learning MRMA Models

The optimization problem defined in Equation 5 is very likely to overfit if we cannot precisely estimate the hyperparameters, which automatically control the generalization capacity of the MRMA model. For instance, $\sigma_U$ and $\sigma_V$ will control the regularization of $U$ and $V$. Therefore, it is more desirable to estimate the parameters and hyperparameters simultaneously during model training. One possible way is to estimate each variable by its maximum a priori (MAP) value while conditioned on the rest variables and then iterate until convergence, which is also known as iterated conditional modes (ICM) [1].

The ICM procedure for maximizing Equation 5 is presented as follows.

**Initialization:** Choose initial values for all variables and parameters.

**ICM Step:** The values of $U$, $V$, $\alpha$ and $\beta$ can be updated by solving the following minimization problems when conditioned on other variables or hyperparameters.

$$
\forall k \in \{1, ..., K\}, \forall i \in \{1, ..., m\} :
$$

$$
U_i^k \leftarrow \arg \min_{U'} \big\{ \frac{1}{2\sigma^2} \sum_{j=1}^{n} \mathbb{1}_{i,j} \big[ \sum_{k=1}^{K} \alpha_i^k \beta_j^k (R_{i,j} - U_i^k (V_j^k)^T)^2 \big] + \frac{1}{2\sigma_U^2} \sum_{k=1}^{K} (U_i^k)^2 \big\},
$$

$$
\alpha_i^k \leftarrow \arg \min_{\alpha'} \big\{ \frac{1}{2\sigma^2} \sum_{j=1}^{n} \mathbb{1}_{i,j} \big[ \sum_{k=1}^{K} \alpha_i^k \beta_j^k (R_{i,j} - U_i^k (V_j^k)^T)^2 \big] + \frac{1}{b_\alpha} \sum_{k=1}^{K} |\alpha_i^k - \mu_\alpha| \big\}.
$$

$$\forall k \in \{1, ..., K\}, \forall j \in \{1, ..., n\}:$$

$$V_j^k \leftarrow \arg\min_{V'} \Big\{ \frac{1}{2\sigma^2} \sum_{i=1}^{m} \mathbb{1}_{i,j} \big[ \sum_{k=1}^{K} \alpha_i^k \beta_j^k (R_{i,j} - U_i^k (V_j^k)^T)^2 \big] + \frac{1}{2\sigma_V^2} \sum_{k=1}^{K} (V_j^k)^2 \Big\},$$

$$\beta_j^k \leftarrow \arg\min_{\beta'} \Big\{ \frac{1}{2\sigma^2} \sum_{i=1}^{m} \mathbb{1}_{i,j} \big[ \sum_{k=1}^{K} \alpha_i^k \beta_j^k (R_{i,j} - U_i^k (V_j^k)^T)^2 \big] + \frac{1}{b_\beta} \sum_{k=1}^{K} |\beta_j^k - \mu_\beta| \Big\}.$$

The hyperparameters can be learned as their maximum likelihood estimates by setting their partial derivatives on $l'$ to 0.

$$\sigma^2 \leftarrow \sum_{i=1}^{m} \sum_{j=1}^{n} \mathbb{1}_{i,j} \big[ \sum_{k=1}^{K} \alpha_i^k \beta_j^k (R_{i,j} - U_i^k (V_j^k)^T)^2 \big] / \sum_{i=1}^{m} \sum_{j=1}^{n} \mathbb{1}_{i,j},$$

$$\sigma_U^2 \leftarrow \sum_{k=1}^{K} \sum_{i=1}^{m} (U_i^k)^2 / Km, \quad \mu_\alpha \leftarrow \sum_{k=1}^{K} \sum_{i=1}^{m} \alpha_i^k / Km, \quad b_\alpha = \sum_{k=1}^{K} \sum_{i=1}^{m} |\alpha_i^k - \mu_\alpha| / Km,$$

$$\sigma_V^2 \leftarrow \sum_{k=1}^{K} \sum_{j=1}^{n} (V_j^k)^2 / Kn, \quad \mu_\beta \leftarrow \sum_{k=1}^{K} \sum_{j=1}^{n} \beta_j^k / Kn, \quad b_\beta = \sum_{k=1}^{K} \sum_{j=1}^{n} |\beta_j^k - \mu_\beta| / Kn.$$

**Repeat:** until convergence or the maximum number of iterations reached.

Note that ICM is sensitive to initial values. Our empirical studies show that setting the initial values of $U^k$ and $V^k$ by solving the classic PMF method can achieve good performance. Regarding $\alpha$ and $\beta$, one of the proper initial values should be $1/\sqrt{K}$ ($K$ denotes the number of sub-models in the mixture model). To improve generalization performance and enable online learning [7], we can update $U, V, \alpha, \beta$ using stochastic gradient descent. Meanwhile, the $\ell_1$ norms in learning $\alpha$ and $\beta$ can be approximated by the smoothed $\ell_1$ method [18]. To deal with massive datasets, we can use the alternating least squares (ALS) method to learn the parameters of the proposed MRMA model, which is amenable to parallelization.

## 6 Experiments

This section presents the experimental results of the proposed MRMA method on three well-known datasets: 1) MovieLens 1M dataset ($\sim$1 million ratings from 6,040 users on 3,706 movies); 2) MovieLens 10M dataset ($\sim$10 million ratings from 69,878 users on 10,677 movies); 3) Netflix Prize dataset ($\sim$100 million ratings from 480,189 users on 17,770 movies). For all accuracy comparisons, we randomly split each dataset into a training set and a test set by the ratio of 9:1. All results are reported by averaging over 5 different splits. The root mean square error (RMSE) is adopted to measure the rating prediction accuracy of different algorithms, which can be computed as follows: $D(\hat{R}) = \sqrt{\sum_i \sum_j \mathbb{1}_{i,j} (R_{i,j} - \hat{R}_{i,j})^2 / \sum_i \sum_j \mathbb{1}_{i,j}}$ ($\mathbb{1}_{i,j}$ indicates that entry $(i,j)$ appears in the test set). The normalized discounted cumulative gain (NDCG) is adopted to measure the item ranking accuracy of different algorithms, which can be computed as follows: $NDCG@N = DCG@N/IDCG@N$ ($DCG@N = \sum_{i=1}^{N} (2^{rel_i} - 1)/\log_2(i+1)$, and $IDCG$ is the $DCG$ value with perfect ranking).

In ICM-based learning, we adopt $\epsilon = 0.00001$ as the convergence threshold and $T = 300$ as the maximum number of iterations. Considering efficiency, we only choose a subset of ranks, *e.g.*, $\{10, 20, 30, ..., 300\}$ rather than $\{1, 2, 3, ..., 300\}$, in MRMA. The parameters of all the compared algorithms are adopted from their original papers because all of them are evaluated on the same datasets.

We compare the recommendation accuracy of MRMA with six matrix approximation-based collaborative filtering algorithms as follows: 1) BPMF [16], which extends the PMF method from a Baysian view and estimates model parameters using a Markov chain Monte Carlo scheme; 2) GSMF [20], which learns user/item features with group sparsity regularization in matrix approximation; 3) LLOR-MA [12], which ensembles the approximations from different submatrices using kernel smoothing; 4) WEMAREC [5], which ensembles different biased matrix approximation models to achieve higher

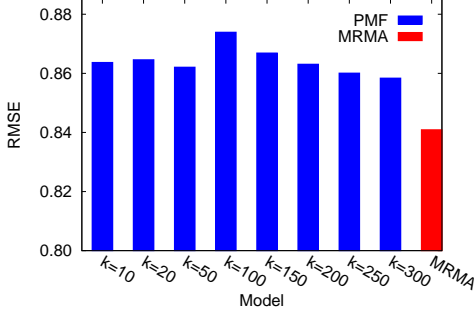 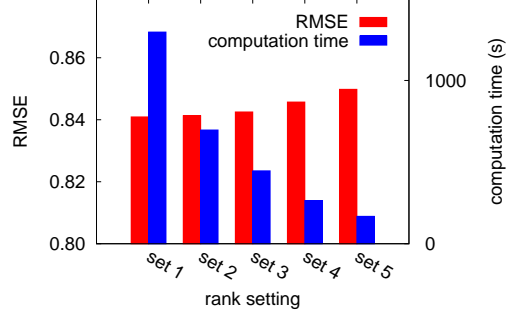

Figure 2: Root mean square error comparison between MRMA and PMF with different ranks.

Figure 3: The accuracy and efficiency tradeoff of MRMA.

accuracy; 5) MPMA [4], which combines local and global matrix approximations using a mixture model; 6) SMA [13], which yields a stable matrix approximation that can achieve good generalization performance.

## 6.1 Mixture-Rank Matrix Approximation vs. Fixed-Rank Matrix Approximation

Given a fixed rank $k$, the corresponding rank-$k$ model in MRMA is identical to probabilistic matrix factorization (PMF) [17]. In this experiment, we compare the recommendation accuracy of MRMA with ranks in $\{10, 20, 50, 100, 150, 200, 250, 300\}$ against those of PMF with fixed ranks on the MovieLens 1M dataset. For PMF, we choose 0.01 as the learning rate, 0.01 as the user feature regularization coefficient, and 0.001 as the item feature regularization coefficient, respectively. The convergence condition is the same as MRMA.

As shown in Figure 2, when the rank increases from 10 to 300, PMF can achieve RMSEs between 0.86 and 0.88. However, the RMSE of MRMA is about 0.84 when mixing all these ranks from 10 to 300. Meanwhile, the accuracy of PMF is not stable when $k \leq 100$. For instance, PMF with $k = 10$ achieves better accuracy than $k = 20$ but worse accuracy than $k = 50$. This is because fixed rank matrix approximation cannot be perfect for all users and items, so that many users and items either underfit or overfit at a fixed rank less than 100. Yet when $k > 100$, only overfitting occurs and PMF achieves consistently better accuracy when $k$ increases, which is because regularization terms can help improve generalization capacity. Nevertheless, PMF with all ranks achieves lower accuracy than MRMA, because individual users/items can give the sub-models with the optimal ranks higher weights in MRMA and thus alleviate underfitting or overfitting.

## 6.2 Sensitivity of Rank in MRMA

In MRMA, the set of ranks decide the performance of the final model. However, it is neither efficient nor necessary to choose all the ranks in $[1, 2, ..., K]$. For instance, a rank-$k$ approximation will be very similar to rank-$(k-1)$ and rank-$(k+1)$ approximations, *i.e.*, they may have overlapping structures. Therefore, a subset of ranks will be sufficient. Figure 3 shows 5 different settings of rank combinations, in which $set\ 1 = \{10, 20, 30, ..., 300\}$, $set\ 2 = \{20, 40, ..., 300\}$, $set\ 3 = \{30, 60, ..., 300\}$, $set\ 4 = \{50, 100, ..., 300\}$, and $set\ 5 = \{100, 200, 300\}$. As shown in this figure, RMSE decreases when more ranks are adopted in MRMA, which is intuitive because more ranks will help users/items better choose the most appropriate components. However, the computation time also increases when more ranks are adopted in MRMA. If a tradeoff between accuracy and efficiency is required, then $set\ 2$ or $set\ 3$ will be desirable because they achieve slightly worse accuracies but significantly less computation overheads.

MRMA only contains three sub-models with different ranks in $set\ 5 = \{100, 200, 300\}$, but it still significantly outperforms PMF with ranks ranging from 10 to 300 in recommendation accuracy (as shown in Figure 2). This further confirms that MRMA can indeed discover the internal mixture-rank structure of the user-item rating matrix and thus achieve better recommendation accuracy due to better approximation.

Table 2: RMSE comparison between MRMA and six state-of-the-art matrix approximation-based collaborative filtering algorithms on MovieLens (10M) and Netflix datasets. Note that MRMA statistically significantly outperforms the other algorithms with 95% confidence level.

| | MovieLens (10M) | Netflix |
|---|---|---|
| BPMF [16] | $0.8197 \pm 0.0004$ | $0.8421 \pm 0.0002$ |
| GSMF [20] | $0.8012 \pm 0.0011$ | $0.8420 \pm 0.0006$ |
| LLORMA [12] | $0.7855 \pm 0.0002$ | $0.8275 \pm 0.0004$ |
| WEMAREC [5] | $0.7775 \pm 0.0007$ | $0.8143 \pm 0.0001$ |
| MPMA [4] | $0.7712 \pm 0.0002$ | $0.8139 \pm 0.0003$ |
| SMA [13] | $0.7682 \pm 0.0003$ | $0.8036 \pm 0.0004$ |
| **MRMA** | $\mathbf{0.7634 \pm 0.0009}$ | $\mathbf{0.7973 \pm 0.0002}$ |

Table 3: NDCG comparison between MRMA and six state-of-the-art matrix approximation-based collaborative filtering algorithms on Movielens (1M) and Movielens (10M) datasets. Note that MRMA statistically significantly outperforms the other algorithms with 95% confidence level.

| Metric | | NDCG@N | | |
|---|---|---|---|---|
| Data \| Method | N=1 | N=5 | N=10 | N=20 |
| Movielens 1M — BPMF | $0.6870 \pm 0.0024$ | $0.6981 \pm 0.0029$ | $0.7525 \pm 0.0009$ | $0.8754 \pm 0.0008$ |
| Movielens 1M — GSMF | $0.6909 \pm 0.0048$ | $0.7031 \pm 0.0023$ | $0.7555 \pm 0.0017$ | $0.8769 \pm 0.0011$ |
| Movielens 1M — LLORMA | $0.7025 \pm 0.0027$ | $0.7101 \pm 0.0005$ | $0.7626 \pm 0.0023$ | $0.8811 \pm 0.0010$ |
| Movielens 1M — WEMAREC | $0.7048 \pm 0.0015$ | $0.7089 \pm 0.0016$ | $0.7617 \pm 0.0041$ | $0.8796 \pm 0.0005$ |
| Movielens 1M — MPMA | $0.7020 \pm 0.0005$ | $0.7114 \pm 0.0018$ | $0.7606 \pm 0.0006$ | $0.8805 \pm 0.0007$ |
| Movielens 1M — SMA | $0.7042 \pm 0.0033$ | $0.7109 \pm 0.0011$ | $0.7607 \pm 0.0008$ | $0.8801 \pm 0.0004$ |
| Movielens 1M — **MRMA** | $\mathbf{0.7153 \pm 0.0027}$ | $\mathbf{0.7182 \pm 0.0005}$ | $\mathbf{0.7672 \pm 0.0013}$ | $\mathbf{0.8837 \pm 0.0004}$ |
| Movielens 10M — BPMF | $0.6563 \pm 0.0005$ | $0.6845 \pm 0.0003$ | $0.7467 \pm 0.0007$ | $0.8691 \pm 0.0002$ |
| Movielens 10M — GSMF | $0.6708 \pm 0.0012$ | $0.6995 \pm 0.0008$ | $0.7566 \pm 0.0017$ | $0.8748 \pm 0.0004$ |
| Movielens 10M — LLORMA | $0.6829 \pm 0.0014$ | $0.7066 \pm 0.0005$ | $0.7632 \pm 0.0004$ | $0.8782 \pm 0.0012$ |
| Movielens 10M — WEMAREC | $0.7013 \pm 0.0003$ | $0.7176 \pm 0.0006$ | $0.7703 \pm 0.0002$ | $0.8824 \pm 0.0006$ |
| Movielens 10M — MPMA | $0.6908 \pm 0.0006$ | $0.7133 \pm 0.0002$ | $0.7680 \pm 0.0001$ | $0.8808 \pm 0.0004$ |
| Movielens 10M — SMA | $0.7002 \pm 0.0006$ | $0.7134 \pm 0.0004$ | $0.7679 \pm 0.0003$ | $0.8809 \pm 0.0002$ |
| Movielens 10M — **MRMA** | $\mathbf{0.7048 \pm 0.0006}$ | $\mathbf{0.7219 \pm 0.0001}$ | $\mathbf{0.7743 \pm 0.0001}$ | $\mathbf{0.8846 \pm 0.0001}$ |

## 6.3 Accuracy Comparison

### 6.3.1 Rating Prediction Comparison

Table 2 compares the rating prediction accuracy between MRMA and six matrix approximation-based collaborative filtering algorithms on MovieLens (10M) and Netflix datasets. Note that among the compared algorithms, BPMF, GSMF, MPMA and SMA are stand-alone algorithms, while LLORMA and WEMAREC are ensemble algorithms. In this experiment, we adopt the set of ranks as $\{10, 20, 50, 100, 150, 200, 250, 300\}$ due to efficiency reason, which means that the accuracy of MRMA should not be optimal. However, as shown in Table 2, MRMA statistically significantly outperforms all the other algorithms with 95% confidence level. The reason is that MRMA can choose different rank values for different users/items, which can achieve not only globally better approximation but also better approximation in terms of individual users or items. This further confirms that mixture-rank structure indeed exists in user-item rating matrices in recommender systems. Thus, it is desirable to adopt mixture-rank matrix approximations rather than fixed-rank matrix approximations for recommendation tasks.

### 6.3.2 Item Ranking Comparison

Table 3 compares the NDCGs of MRMA with the other six state-of-the-art matrix approximation-based collaborative filtering algorithms on Movielens (1M) and Movielens (10M) datasets. Note that for each dataset, we keep 20 ratings in the test set for each user and remove users with less than 5

ratings in the training set. As shown in the results, MRMA can also achieve higher item ranking accuracy than the other compared algorithms thanks to the capability of better capturing the internal mixture-rank structures of the user-item rating matrices. This experiment demonstrates that MRMA can not only provide accurate rating prediction but also achieve accurate item ranking for each user.

## 6.4 Interpretation of MRMA

Table 4: Top 10 movies with largest $\beta$ values for sub-models with rank $k = 20$ and $k = 200$ in MRMA. Here, #ratings stands for the average number of ratings in the training set for the corresponding movies.

| rank=20 | | | rank=200 | | |
|---|---|---|---|---|---|
| movie name | $\beta$ | #ratings | movie name | $\beta$ | #ratings |
| Smashing Time | 0.6114 | | American Beauty | 0.9219 | |
| Gate of Heavenly Peace | 0.6101 | | Groundhog Day | 0.9146 | |
| Man of the Century | 0.6079 | | Fargo | 0.8779 | |
| Mamma Roma | 0.6071 | | Face/Off | 0.8693 | |
| Dry Cleaning | 0.6071 | 2.4 | 2001: A Space Odyssey | 0.8608 | 1781.4 |
| Dear Jesse | 0.6063 | | Shakespeare in Love | 0.8553 | |
| Skipped Parts | 0.6057 | | Saving Private Ryan | 0.8480 | |
| The Hour of the Pig | 0.6055 | | The Fugitive | 0.8404 | |
| Inheritors | 0.6042 | | Braveheart | 0.8247 | |
| Dangerous Game | 0.6034 | | Fight Club | 0.8153 | |

To better understand how users/items weigh different sub-models in the mixture model of MRMA, we present the top 10 movies which have largest $\beta$ values for sub-models with rank=20 and rank=200, show their $\beta$ values, and compare their average numbers of ratings in the training set in Table 4. Intuitively, the movies with more ratings (*e.g.*, over 1000 ratings) should weigh higher towards more complex models, and the movies with less ratings (*e.g.*, under 10 ratings) should weigh higher towards simpler models in MRMA.

As shown in Table 4, the top 10 movies with largest $\beta$ values for the sub-model with rank 20 have only 2.4 ratings on average in the training set. On the contrary, the top 10 movies with largest $\beta$ values for the sub-model with rank 200 have 1781.4 ratings on average in the training set, and meanwhile these movies are very popular and most of them are Oscar winners. This confirms our previous claim that MRMA can indeed weigh more complex models (*e.g.*, rank=200) higher for movies with more ratings to prevent underfitting, and weigh less complex models (*e.g.*, rank=20) higher for the movies with less ratings to prevent overfitting. A similar phenomenon has also been observed from users with different $\alpha$ values, and we omit the results due to space limit.

## 7 Conclusion and Future Work

This paper proposes a mixture-rank matrix approximation (MRMA) method, which describes user-item ratings using a mixture of low-rank matrix approximation models with different ranks to achieve better approximation and thus better recommendation accuracy. An ICM-based learning algorithm is proposed to handle the non-convex optimization problem pertaining to MRMA. The experimental results on MovieLens and Netflix datasets demonstrate that MRMA can achieve better accuracy than six state-of-the-art matrix approximation-based collaborative filtering methods, further pushing the frontier of recommender systems. One of the possible extensions of this work is to incorporate other inference methods into learning the MRMA model, *e.g.*, variational inference [8], because ICM may be trapped in local maxima and therefore cannot achieve global maxima without properly chosen initial values.

## Acknowledgement

This work was supported in part by the National Natural Science Foundation of China under Grant No. 61332008 and NSAF under Grant No. U1630115.

## Footnotes

*This work was conducted while the author was with IBM.

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
