[Reviews · NeurIPS 2017]

Reviewer 1



This paper proposed a mixture rank matrix factorization which decompose a low rank matrix as a mixture of sub-matrices with low-rank. The proposed approach is an extension of probabilistic matrix factorization and it's been shown that it has a superior performance compare to existing methods. I have the following concern: 1- Paper presentation is not very well and it includes all the derivation for optimization algorithm which could move to the appendix and instead some analysis regarding the convergence and computational complexity could be added to the main text. 2- This is not quite clear from the text that how the model perform differently if the submatrices have overlap structure. I guess in this case there would be a lot of scenarios and computationally make it intractable. 3- By imposing more structure in the matrix, there are more hyperparamters to be optimized and looking at the experimental results, in most cases the improvement is very marginal and not convincing.

Reviewer 2



This is an excellent paper, proposing a sound idea of approximating a partially defined rating matrix with a combination of multiple low rank matrices of different ranks in order to learn well the head user/item pairs (users and items with lots of ratings) as well as the tail user/item pairs (users and items we few ratings). The idea is introduced clearly. The paper makes a good review of the state-of-the-art, and the experiment section is solid with very convincing results. In reading the introduction, the reader could find controversial the statement in lines 25-27 about the correlation between the number of user-item ratings and the desired rank. One could imagine that a subgroup of users and items have a large number of ratings but in a consistent way, which can be explained with a low rank matrix. The idea is getting clear further in the paper, when explained in the light of overfitting and underfitting. The ambiguity could be avoided in this early section by adding a comment along the line of “seeking a low rank is a form of regularization”.